# Management of residents in difficulty in a Swiss general internal medicine outpatient clinic: Change is necessary!

Cédric Lanier[1]☯*, Virginie Muller-Juge[1]☯, Melissa Dominicé Dao[2], Jean-Michel Gaspoz[3], Noëlle Junod Perron[4], Marie-Claude Audétat[1,4]☯

1 Family Medicine Unit (UIGP), University of Geneva, Geneva, Switzerland, 2 Department of Community Medicine, Primary Care and Emergency Medicine, Geneva University Hospitals, Geneva, Switzerland, 3 Hirslanden Clinique des Grangettes, Chênes-Bougeries, Switzerland, 4 Unit of Development and Research in Medical Education (UDREM), University of Geneva, Geneva, Switzerland

☯ These authors contributed equally to this work.
* cedric.lanier@unige.ch

## Abstract

### Aims of the study

Residents in difficulty are a major cause for concern in medical education, with a prevalence of 7–15%. They are often detected late in their training and cannot make use of remediation plans. Nowadays, most training hospitals in Switzerland do not have a specific program to identify and manage residents in difficulty. The aim of the study was to explore the challenges perceived by physicians regarding the process of identifying, diagnosing, and supporting residents in difficulty in a structured and programmatic way. We explored perceptions of physicians at different hierarchical levels (residents (R), Chief residents (CR), attending physicians (A), Chief Physician (CP)) in order to better understand these challenges.

### Methods

We conducted an exploratory qualitative study between December 2015 and July 2016. We asked volunteers from the Primary Care Division of the Geneva University Hospitals to partake to three focus groups (with CR, A, R) and one interview with the division's CP. We transcribed, coded, and qualitatively analyzed the three focus groups and the interview, using a content thematic approach and Fishbein's conceptual framework.

### Results

We identified similarities and differences in the challenges of the management of residents in difficulty on a programmatic way amongst physicians of different hierarchical levels.
Our main findings:

• Supervisors (CR, A, CP) have good identification skills of residents in difficulty, but they did not put in place systematic remediation strategies.

**Data Availability Statement:** There are ethical restrictions on sharing the data publicly, as it contains sensitive information from doctors and could facilitate participant identification. Certain

topics are sensitive on an institutional level and can only be accessed upon request through the University of Geneva's Yareta repository (https://doi.org/10.26037/yareta:d5unipemxjahvno53oyla4flqu). Alternatively, data access requests may be sent to the ethics committee IRB of the University of Geneva - UNIGE and the IRB of the university hospitals - HUG - CCER (Sophie.Desjacques@unige.ch).

**Funding:** The funders had no role in study design, data collection and analysis, decision to publish, or preparation of the manuscript. This project was supported by institutional Funding Elie SAFRA from the Faculty of Medicine, University of Geneva.

**Competing interests:** The authors have declared that no competing interests exist.

- Supervisors (CR, A) were concerned about managing residents in difficulty. They were aware of the possible adverse effects on patient care, but "feared to harm" resident's career by documenting a poor institutional assessment.

- Residents "feared to share" their own difficulties with their supervisors. They thought that it would impact their career negatively.

- The four physician's hierarchical level reported environmental constraints (lack of funding, time constraint, lack of time and resources. . .).

## Conclusion

Our results add two perspectives to specialized recommendations regarding the implementation of remediation programs for residents in difficulty. The first revolves around the need to identify and fully understand not only the beliefs but also the implicit norms and the feeling of self-efficacy that are shared by teachers and that are likely to motivate them to engage in the management of residents in difficulty. The second emphasizes the importance of analyzing these elements that constitute the context for a change and of identifying, in close contact with the heads of the institutions, which factors may favor or hinder it. This research action process has fostered awareness and discussions at different levels. Since then, various actions and processes have been put in place at the Faculty of Medicine in Geneva.

## Introduction

A resident in difficulty is defined as a resident who "does not meet the expectations of the training program because of a significant problem with knowledge, attitudes or skills" [1]. These residents are a major cause for concern in medical education and are often detected too late in their training [2–4]. Residents in difficulty are a common issue with a prevalence of 7–15% [5, 6]. If we refer to the widely applied Vaughn's conceptual framework [7], they may experience four types of difficulties: 1) Cognitive (i.e. lack of knowledge, clinical reasoning difficulties); 2) Affective (i.e. low self-esteem, anxiety. . .); 3) Structural (i.e. disorganization, poor time management. . .); 4) Interpersonal (i.e. lack of professionalism, inappropriate behavior towards patients and colleagues. . .). Several types of difficulties are often present simultaneously [8, 9].

Recommendations to manage residents in difficulty include implementing a remediation program [8, 10]. Remediation programs rely on an organizational structure, supported by local authorities, which allows early identification of residents in difficulty through a programmatic (step by step) approach or procedure. The different steps of this procedure include: documenting the resident's difficulties, formulating a pedagogical diagnosis, and elaborating a targeted and individualized and targeted learning plan. This program must define specific objectives, and consequences in case of failure of remediation [11–16]. Remediation programs appear to have good results with 75–90% of successful remediation [9, 17, 18].

Most of the remediation programs studied were conducted in North American countries, where postgraduate training programs are very well structured and include formal assessment programs. Such programs can be either traditional, with module-test building blocks at the end of each rotation, or programmatic. Programmatic assessment involves continual collection and analysis of routine information about the learner's competence and progress and may be supplemented by purposively collected additional assessment information. This approach offers a maximal amount of information both for the learners and their supervisors, and

justifies the high-stakes decisions made at the end of a training phase [19]. Both systems require the achievement of different rotations in order to pass the certification exam [10, 17, 20]. In other countries, postgraduate training programs are less structured and include looser assessment programs. For example, in Switzerland where postgraduate training lasts 5 to 6 years according to the medical discipline, the requirements for obtaining a board certification of specialist are the following: 1) a yearly certificate established by the head of the postgraduate training center, acknowledging that the candidate demonstrates the expected skills; 2) four yearly workplace-based assessments that take the form of mini CEX or DOPS [21]; 3) a log-book documenting the necessary technical procedures for each discipline; 4) a final written and knowledge/problem-solving based exam using multiple choice questions [21]. According to our experience, the percentage of residents in difficulty in Switzerland are quite similar to those mentioned in published studies. However, heads of divisions rarely establish yearly certificates reporting candidates' insufficient skills. Furthermore, there are no national requirements for remediation processes regarding residents in difficulty. Hence, clinical educators may feel at loss regarding the management of residents in difficulties [15, 16, 22, 23].

In this context, our research aimed to obtain an overview of the state of practices in the largest Swiss training center in ambulatory general internal medicine and to understand how these practices are perceived. We also explored the feasibility of implementing a remediation program for residents in difficulty in such as a Swiss division.

## Materials and methods

### Design

We conducted a participation action research [24] in the Primary Care Division of the Geneva University Hospitals. This research methodology is recommended when implementing changes [25, 26], and aims to actively involve and include participants in the process of change. In this perspective, we conducted our study through focus groups, as well as one-to-one semi structured interview [27].

As several hierarchical levels are usually involved in the supervision of the residents, we chose to deepen our analysis with a cross sectional approach to identify potential differences between the different stakeholders in regard to their attitudes or normative beliefs. We believed that this would help to identify specific implementation actions for each hierarchical level, in order to offer a coherent approach of residents in difficulty. Other studies in medical education have used this multi-level approach to understand different levels' perceptions of specific issues [28].

We used two different conceptual frameworks. The first framework developed by Fishbein [29, 30], is a predictive model of behavior that allows to understand the different factors influencing the intention of performing (or not) a specific behavior (in our case, the follow up of struggling residents) [31, 32]. As illustrated in Fig 1, Fishbein's framework postulates that a specific behavior depends on the individual's skills, on environmental constraints and on the individual's intention to act. The latter depends on the individual's beliefs, norms and feelings of self-efficacy regarding this behavior [29].

The second framework assessed local practices and their relative strengths, weaknesses, opportunities, and threats (SWOT analysis) before designing a concrete plan for implementing a remediation program for residents in difficulty [33].

### Context and participants

We conducted our study in the Division of Primary Care Medicine of the Department of Community Medicine, Primary Care and Emergency Medicine at the University Hospitals of Geneva, in Switzerland.

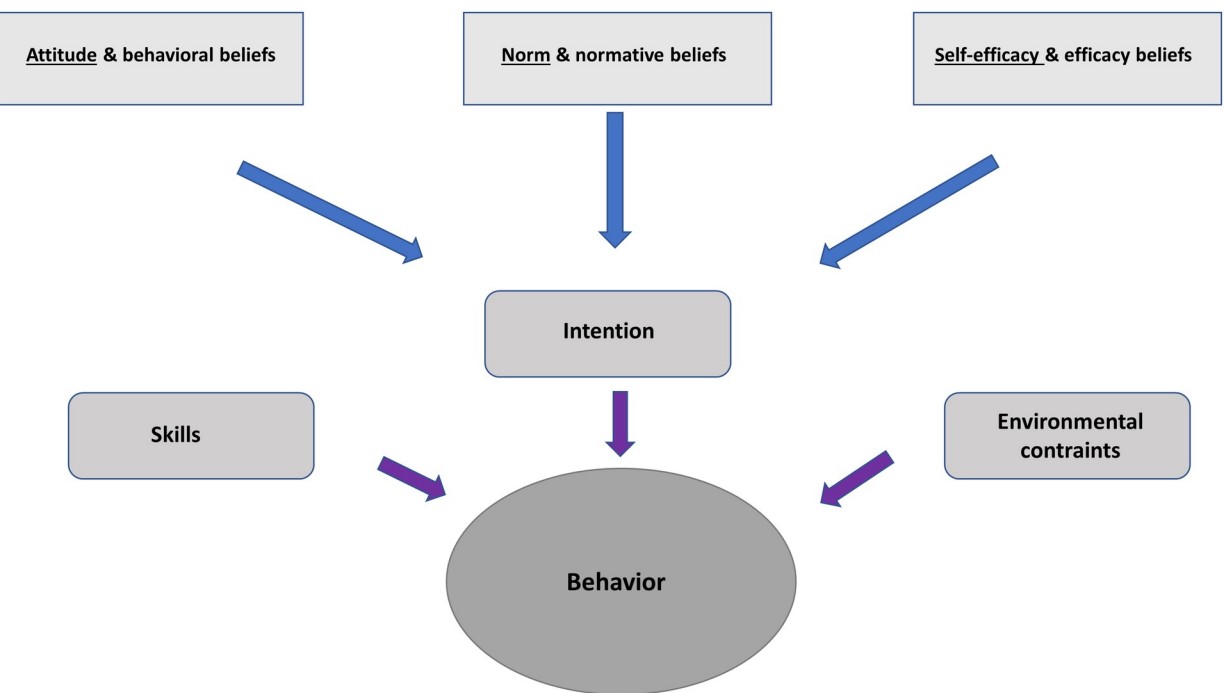

**Fig 1. Fishbein's integrative model of behavior prediction (adapted from [29]).**

The mission of the Division of Primary Care Medicine is to deliver medical care, to provide pre- and post-graduate training, and to conduct research in community-based clinics in Geneva, a province of 450'000 inhabitants. It provides 38'000 planned medical consultations each year in addition to 23'000 emergency consultations at the medical and surgical walk-in clinics every year. Its patient population includes residents of the Geneva region, with a high percentage of vulnerable populations, such as asylum seekers, undocumented migrants, patients without insurance coverage, prisoners, frail elderly people bound to their homes or patients with substance abuse. At the time of our study, the division trained 60 residents (R) in ambulatory care, directly supervised by 33 Chief residents (CR), depending on the organization of the different units of the Division. (Chiefs residents have completed their postgraduate training, and are in charge of the clinical consultations and daily supervision in the Division) [34]. The 33 Chief residents were under the responsibility of 12 attending physicians (A). In this article, we will use the term supervisors for direct supervisors, regardless of, if they are Chief residents (CR) or attending physicians (A). Residents (R) spend 12 to 24 months training in the Primary Care Division at the end of a five-year general internal medicine residency training.

## Ethics statement

The study was approved by the ethics committee of the State of Geneva Institutional Review Board. Participation was voluntary and participants signed an informed consent form.

## Data collection

We conducted three focus groups (range 80–83 minutes) between December 2015 and April 2016, with respectively residents (not identified as in difficulty), Chief residents, attending physicians and one interview with the Chief physician (division chief) (41 minutes).

Distinguishing between these different levels seemed important to us to facilitate communication between participants and researchers [27], but also to take into account previous research results having underlined that students are mainly influenced during supervision by residents and chiefs residents (70%) rather than by experienced physicians [34].

We used a purposeful sampling strategy for the recruitment. Following the action research's principles, our aim was to involve participants interested in becoming partners in the research process. In this perspective, we wanted a representative sample of supervisors (attending physicians, Chief residents) involved in direct supervision of residents. Potential participants were informed by e-mail of the nature and objectives of the study and assured of confidentiality.

We chose residents that were not in difficulty, as it would have been too stigmatizing to interview residents in difficulty. We also interviewed the Chief physician to obtain the perception of the top of the hierarchy; Chief physician was interviewed separately to allow other participants to freely express their thoughts and potential difficulties encountered [27]. This also allowed the Chief physician to talk openly about his issues in relation to other departments or hospital requests. The focus groups and the interview were conducted by members of the study, external to the Division team (CL, VMJ, MCA,), respectively physician, research assistant and professor, specialist in medical education). The focus groups and the interview had nine questions to explore the five main themes described by Fishbein (attitude beliefs, normative beliefs, self-efficacy beliefs, skills, environmental constraints) relative to the management of residents in difficulty [29] (See Supplementary material). Before each focus group session, the facilitator repeated the information and obtained written consent. A pilot study was done to test out our semi-structured interview guide and to ensure that our questions would be sufficiently precise and understandable.

Data were anonymized during transcription. All the data were stored outside the division in a locked office.

## Data analysis

Preliminary analysis was conducted by the three main researchers (CL, VMJ, MCA), using the conceptual frameworks described above [29, 35], and following a theory-driven variant of immersion–crystallization [36]. Theory-driven immersion—crystallization not only recognizes that data interpretation cannot be entirely free from the researchers' perspectives, but argues that inductive (data-driven) and deductive (theory-driven) processes should be brought together to enrich interpretation [36]. Disagreements were resolved by consensus and allowed further refinement of the coding scheme.

First, two researchers (CL, VMJ) independently identified, coded, and classified the content of the transcripts using the ATLAS.ti software. They then compared and discussed their coding; disagreements were resolved by consensus and allowed further refinement of the coding scheme. The third researcher (MCA) cross-checked the coding.

The coding researchers then sought to corroborate their findings by providing their co-authors, who were members of the Primary care Division, with the results of their analysis and asking them whether the findings were consistent with their experience within the Division [37].

## Results

### Sociodemographic characteristics

Three focus groups were conducted with respectively 6 residents (R), 7 Chief residents (CR), and 7 attending physicians (A). One individual interview took place with the Chief Physician (CP). The sociodemographic characteristics of the participants are described in Table 1.

**Table 1. Socio-demographic characteristics of the participants.**

|  | Participants | Gender (n; %) | Age (mean;) | Supervision experience (yrs) (mean;) | Training skills yes (n; %) |
|---|---|---|---|---|---|
| FG 1 | 6 R | Male: 3 (50) | 34 | NA | NA |
|  |  | Female: 3 (50) |  |  |  |
| FG 2 | 7 CR | Male: 2 (28,6) | 36.7 | 1.4 | 6 (85,7) |
|  |  | Female: 5(71.4) |  |  |  |
| FG 3 | 7 A | Male: 5 (71.4) | 47,1 | 11.6 | 4 (57.1) |
|  |  | Female: 2 (28.6) |  |  |  |
| ITW | 1 CP | Male | 63 | 30 | No |

FG = Focus Group; ITW = Interview; R = Resident; CR = Chief Resident; A = Attending Physicians; CP = ChiefPhysician; Training skills = completed supervision workshops or a Certificate of advanced studies in medical education.

The following results are organized according to the different topics of Fishbein's conceptual framework. We first provided a definition of the category, followed by findings and quotes from our data regarding this category. We also classified our results according to their favorable or unfavorable potentialities, in order to take into account a SWOT analysis (Strengths, Weaknesses, Opportunities, Threats), regarding future implementation of a remediation program for residents in difficulty [33].

## Attitude and behavioral beliefs toward the management of residents in difficulty: Willing supervisors undermined by resistant residents

This category is related to the individual's attitudes and beliefs about the management of residents in difficulty. These beliefs can influence the intention to support or not residents in difficulty (i.e does the supervisors think that remediation programs are effective and useful?).

We observed a general favorable attitude of supervisors towards residents in difficulty, as well as a genuine desire to help them.

*CR2 (FG2, line 834): "I have the impression that there's a real awareness of this problem and a strong desire to take care of this problem, to detect them earlier".*

Participants also expressed a feeling of social responsibility to train competent physicians. They were aware of the future risk for patients if incompetent physicians were allowed to practice without supervision after achieving their post-graduate training.

*CP (ITW, line 79): "In an earlier position as an attending physician (. . .) I encountered residents in difficulty and often Chief residents wanted to get rid of them. (. . .) My boss at the time said: "it's very dangerous to do this because these people will eventually set up private practices and so we will be putting these dangerous people "on the market".*

Supervisors were aware of the importance of a programmatic approach to manage residents in difficulty and mentioned several aspects in line with the international recommendations [14]. These included the necessity to define clear requirements towards residents, to separate evaluation and supervision roles and to work in teams to supervise, share and manage residents in difficulty. Finally, supervisors raised the issue that managing residents in difficulty was time consuming, but they were aware of the need to be trained to become a competent supervisor.

Despite these favorable attitudes, supervisors held the belief that the success of a remediation program depended on the resident's attitude and his willingness to engage in the process. They mentioned how some residents contested a negative evaluation or their supervisor's advice, making the supervisor feel powerless to handle the situation.

*CR 6 (FG2, line 674): "As my colleagues mentioned, I believe that the efficacy [of the remediation] depends on how receptive the resident is. It can be very efficient if the resident accepts it, but if there is a denial, the efficacy could be very bad".*

These beliefs were shared by all supervisors, whatever their hierarchical position.

The supervisors' favorable attitude and beliefs about the management of residents in difficulty collide with a number of perceived norms and normative beliefs mentioned below.

## Norm and normative beliefs: Lack of formal structure and fear of stigmatization

This category is related to normative beliefs, which supervisors will refer to regarding the management of residents in difficulty. In this perspective, supervisors will behave based on what others would do and which behaviors are expected of them, i.e. in line with the norm and implicit normative beliefs present within their workplace.

Supervisors mentioned that there was no defined process to manage residents in difficulty within the division regarding the evaluation, supervision and remediation process. Furthermore, there was no consensus on formal requirements towards residents inside the division.

*CR7 (FG2, line 323):" I think that (. . .) [it's necessary] to standardize requirements . . . If there are division requirements (..), that [it's clear that] this is what we are supposed to do and that it is required for everybody, it's maybe easier to bring it on".*

In addition, there was no clear distribution of the different tasks between supervisors regarding the management of residents in difficulty. Thus, the level of involvement of the different supervisors (Chief residents, attending physicians) varied depending on the person and the situation. Some Chief residents expressed a need to be supported by the attending physician when managing residents in difficulty. Some administrative aspects of the evaluation process were also challenging. For instance, the Chief physician had to justify himself before the central committee of the Swiss postgraduate body in case he did not validate a resident's rotation. This discouraged supervisors not to validate a rotation for a resident in difficulty, in line with the resistance coming from the hierarchy and the division of human resources.

Specific inhibiting normative beliefs were found at each level of hierarchy.

*The residents* reported a taboo when it came to expressing that they were in difficulty. For them, it meant weakness and they were afraid of being stigmatized. This made them reluctant to talk to their colleagues or supervisors about it.

*R5 (FG1, line 312): "With colleagues, it's like, it's more difficult to talk about it. I think that it is a bit taboo. It would in fact be like revealing that we are weak".*

Residents also trivialized their difficulties and felt that all residents encountered some level of difficulties. They related these difficulties to excessive workload and were not aware that colleagues could have cognitive or other types of difficulties.

*Chief residents and attending physicians* conceptualized supervision as mostly a process of supporting residents rather than a process that included evaluation and remediation. They

were particularly concerned not to be judgmental of the resident and were afraid to harm residents in difficulty and their career by providing them with a poor evaluation.

*CR4 (FG2, line 229): "Again, for the final evaluation, often we recognize the resident's difficulties, but at the end of the year, we often have a hard time writing things as they truly are".*

*The Chief physician* mentioned a strong financial performance norm. Indeed, the institution asks all divisions to obtain a positive financial performance. A resident, if identified as being in difficulty, would benefit from a reduced clinical activity and would need more supervision time, thus impacting negatively on the financial performance of the division and causing potential tensions between the chief physician and the institution's top management.

*CP (ITW, line 408): "They put considerable pressure. . . every three months I have a reporting session with the hospital direction. I can tell you that they look at the numbers. . . every three months, I have to justify these statistics in front of the direction of the hospital".*

This interview, as well as the different focus groups, emphasized the absence of a remediation culture inside the larger institution. There was no institutional incentive or formal policy prescribed by the training hospital to manage residents in difficulty.

*CP (ITW, line 292): "What must be said is that the idea of managing these residents in difficulty is, well it doesn't exist in this hospital, so we are pioneers with this idea".*

These inhibiting norms and normative beliefs probably caused uncertainty for the supervisors' and impacted their sense of self-efficacy as described in the next paragraph.

## Self-efficacy and efficacy beliefs: Supervisors in difficulty!

This category relates to how the supervisors perceive their own skills and efficacy in managing residents in difficulty (i.e does the supervisor feel competent to manage residents in difficulty?) All supervisors reported a good sense of self-efficacy when identifying residents in difficulty. However, their identification process was mostly intuitive. Supervisors mentioned having difficulties to further evaluate residents in difficulty more rigorously and to establish a pedagogical diagnosis. They needed validation from their peers and wanted specific tools to help them assess and confront residents, especially when performance was considered weak.

*CR7 (FG2, line 417): "So, I have the impression that in the end, when there is a difficulty, for me to endorse the poor evaluation, I need the support of my peers, their validation".*

Supervisors also mentioned conflicting roles when managing residents in difficulty. They reported feeling uneasy and divided in situations where they had to endorse multiples roles simultaneously: i.e. the roles of caregiver, supervisor, and evaluator to the resident in difficulty. They felt the need to act as caregiver for residents perceived as distressed but recognized that it was in contradiction with their role as supervisors who must give a negative evaluation regarding the same resident's lower performance. Indeed, helping the resident to improve and, at the same time, being the evaluator who will decide if the resident passes or fails his rotation was perceived as discordant.

*A5 (FG3, line 552):* ... *"We believe that our asset is to be a doctor, so we don't stay in our role of team manager, but we become the treating physician of our resident, because we know how to do it, although we should not do it".*

*CR3 (FG2, line 544):* *"And we have this double role: on one hand we are the kind supervisor who checks out how the resident is doing, and then at some point we have a hierarchical role and we must say « This is how it is ». And sometimes, it's just, I find it difficult to balance".*

In summary, the supervisor's lack of confidence in their self-efficacy when faced with a resident in difficulty resulted in the supervisor himself feeling in difficulty, through a mirroring effect. Furthermore, some supervisors were not convinced of the efficacy of remediation process for residents in difficulty.

*CR4 (FG2, line 431):* *"And then, somehow, the resident sends back to us an image of us being a lousy supervisor, in the sense that we question our competence. We tell ourselves: „It's me who can't do it. Is it my fault maybe if the resident isn't progressing"?*

This was more often mentioned by supervisors directly in contact with residents (attending physicians, Chief residents), but also expressed by attending physicians not supervising residents directly.

This feeling of uncertainty regarding their self-efficacy seemed to be related to the insufficient pedagogical skills held by supervisors, as described below.

## Supervisors' awareness of their patchy skills to manage residents in difficulty

This category is related to the necessary skills required of the supervisor for the management of residents in difficulty. The skills mentioned were not necessarily mastered by supervisors but were mentioned as useful.

They felt able to detect symptoms and signs of residents in difficulty. Some supervisors had also received feedback training and were aware of how these skills helped them to manage residents in difficulty.

*A2 (FG3, line 75):* *"We observe how they (residents) manage their patients. So, we can analyze that ... there might be some shortcomings in the diagnostic approach, in the approach to treatment, and well, that's more or less it".*

However, Chief residents and attending physicians expressed their lack of specific pedagogical skills in regard to the implementation of a support plan, monitoring progress, according to defined pedagogical objectives. They also mentioned that they needed pedagogical tools to manage residents in difficulty (i.e assessment tools).

*A2 (FG3, line 545):* *"I would add the, in the range of competencies, pedagogical competencies, meaning how to formulate one or several objectives, because at some point we must lay down individualized learning objectives for these people. (. . .) And exactly that: how to set an objective, how to evaluate it afterwards. That is, I think, it entails how we give the feedback in the follow up in fact".*

Finally, as proposed by Fishbein's conceptual framework, we analyzed constraints of the local setting that could influence the implementation of a remediation program for residents in difficulty.

### Environmental constraints: Time constraint and funding undermine remediation

This category relates to external aspects that will impact on how the individual can follow through with the intended behavior.

In continuum of the inhibiting institutional norms previously mentioned, the attending physicians observed that the institution did not allocate any specific resources (people and time) to manage residents in difficulty. This resulted in a lack of continuity in the supervision process.

*A5 (FG3, line 976): "I realize that, in my team, these past two years, even if we wanted to do it, there we times during the year when I wasn't capable of doing it because we did not have the resources to do so".*

Moreover, the Chief physician underlined that the training hospital was not under any pressure from the Swiss postgraduate medical training bodies to focus on the management of residents in difficulty. Rather, the hospital was involved and giving priority to other major projects.

*CP (ITW, line 604): "The problem is that it's a project that, well, we will have to advertise it as a project, but the general direction of the hospital is not under any external constraint that will impose it upon them, for the time being".*

The Fig 2 summarizes our results and organizes them according to their favorable or unfavorable impact on possible implementation of a remediation program for residents in difficulty.

## Discussion

### Summary of the results

The aim of this research was to shed some light on the local perceptions and practices about the management of residents in difficulty, and to study the feasibility of implementing a remediation program for residents in difficulty in a Swiss division of ambulatory general internal medicine.

Most of our results were similar with those observed in literature. According to Fishbein's categories, we observed a positive and willingness from supervisors to manage residents in difficulty, with a strong feeling of social responsibility to train competent physicians [23]. But we also identified many barriers that could interfere with the future implementation of a remediation program for residents in difficulty. We observed several inhibiting normative beliefs, i.e the fear to harm the resident and the fear of conflict [23, 31]. Supervisors felt able to identify residents in difficulty but had poor feelings of self-efficacy to evaluate and manage them [23, 38, 39]. All hierarchical levels (Chief Physician, attending physicians, Chief residents, residents) reported a lack of resources and time, leading to a lack of continuity in the supervision processes [8, 23, 31, 40].

We observed a similar narrative at each hierarchical levels of physicians (Chief Physician, attending physicians, Chief residents, and residents), except for aspects related to supervisors' experience and responsibilities. For example, Chief residents felt more uncertain than the attending physicians regarding their self-efficacy, and the Chief Physician and attending physicians were more concerned by institutional and organizational aspects.

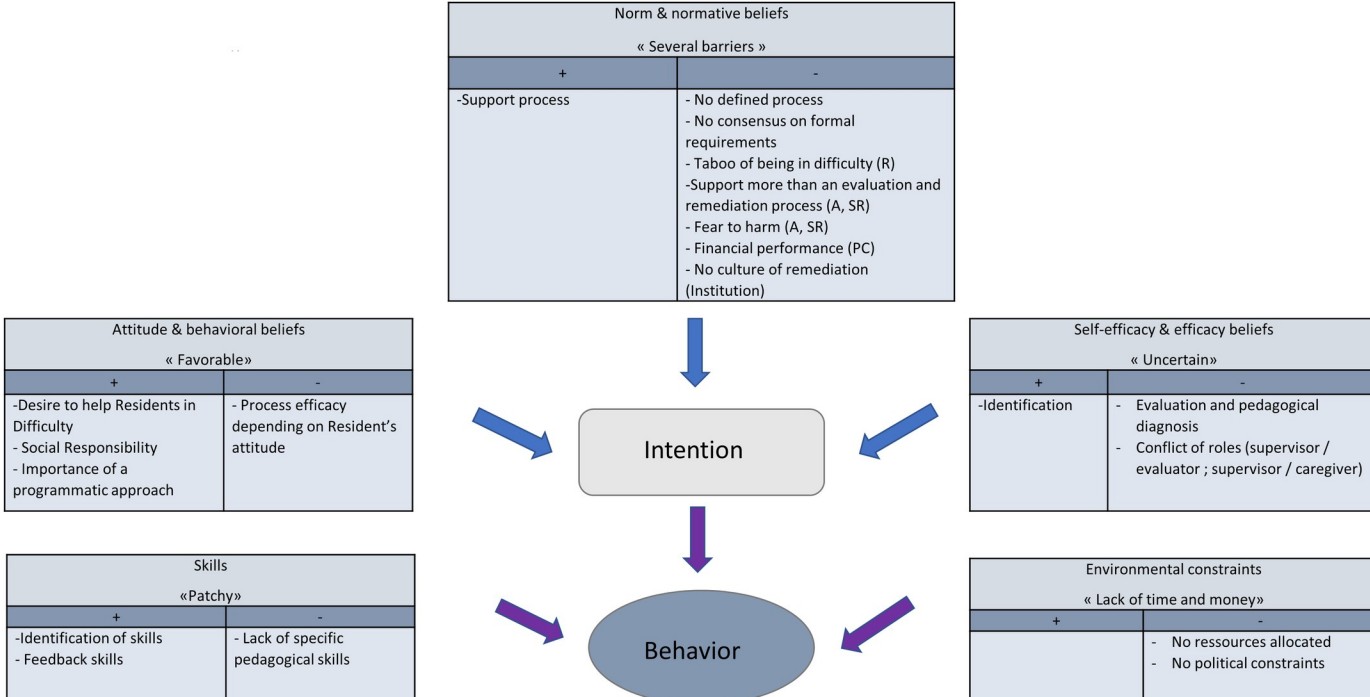

**Fig 2. Summary of results integrated in the Fishbein's conceptual framework and classified by their favorable or unfavorable potentialities regarding a future implementation of a remediation program for residents in difficulty.**

Our results are in line with specialized literature in medical education regarding these issues, but our in-depth analysis allowed us to fully understand challenges in place with a perspective of a change their practice.

Already used in medical education for research on similar topics [31], the Fishbein's integrative model of behavioural prediction allowed us to underpin our data collection and analysis.

## Bringing forward two paradoxes

The synthesis of our results through the Fishbein conceptual framework [29] revealed two main paradoxes that hinder potential changes.

**1. Supervisor as caregiver or teacher?.** The first paradox relates to the tension between a relational approach versus a pedagogical approach for supervisors managing residents in difficulty. The medical norm of "*do no harm*" results mainly in supervisors being kind and understanding with their residents in difficulty. This aspect, combined with the lack of specific pedagogical skills, inhibit supervisors to efficiently tackle the residents' difficulties. Attending physicians and Chief residents were ill at ease handling this double role of supervisor/evaluator and caregiver, and assuming these potentially conflicting responsibilities when faced with a resident in difficulty. As observed by Earle-Foley et al., the assessment "could be viewed as uncaring behavior" and be in tension with the care for others related to the profession [41].

Supervisors viewed the remediation process as depending strongly on the residents in difficulty's favorable attitudes. Specific training for supervisors could help them to feel more proactive and more confident with the measures they implement for a resident in difficulty [10]. Furthermore, changing the Swiss postgraduate curriculum to base evaluation on outcomes rather than time spent in clinical rotations would be beneficial. To date, this change has

already been made in the undergraduate curriculum [42]. A better alignment between the evaluation methods and the outcomes would make the pedagogical approach more explicit and legitimate, and less dependent on residents' and supervisors' attitudes [43, 44]. This change would also be in line with the evolution of medical education towards a learner-centered approach, as it is progressively implemented in a wide number of training institutions.

**2. Supervisor as teacher or institutional employee?.**  We observed a second paradox between the residents' training needs versus the institutional performance needs. Although institutions need to employ well-trained physicians, they also need to treat a minimal number of patients with the allocated financial resources. Thus, supervisors' motivation to manage residents in difficulty is in tension with the financial and "clinical volume" requirements imposed by the institution. This underlines the strong influence of institutions' professional needs over the postgraduate training of residents in Switzerland, where residents are expected to run the institutions. Post graduate training is not under the responsibility of the University, and hospitals do not have the same priorities as the University. For instance, hospitals may not have as a priority to develop a remediation culture and to implement systematic processes for the management of residents in difficulty. In Switzerland, supervisors' written evaluations are the only direct link between the hospitals and the Swiss Institute for Advanced Medical Training and Education (SIWF) that validate the postgraduate training [45]. This autonomic institute is linked to the Swiss Medical Association (FMH) [46] and has the duty to organize postgraduate training and to validate postgraduate titles [45]. This is in contrast with most studied remediation programs for residents in difficulty, where the University has the social responsibility to train competent physicians [10, 14, 17].

In summary, our results suggest that all stakeholders involved the resident evaluation process did not feel legitimate to formally evaluate residents in difficulty for various reasons: residents were reluctant to acknowledge their difficulties; chief residents felt isolated and powerless in evaluating struggling resident and would not report poor performance to protect the residents; attending physicians could not rely on institutional systematic process to start a remediation program; the division chief did not have additional financial resources to manage a resident in difficulty.

It is likely that as long as not only the supervisors, but also the institution remains muddled between these two paradoxes, implementation of a remediation program cannot be successful.

Since supervisors are aware of their inconsistent skills to manage residents in difficulty, developing their pedagogical skills may allow them to feel more empowered. This should go together with the development of their professional identity [47]: what it means to become a teacher and how new explicit and implicit norms are acquired and put into action. Furthermore, because of the two paradoxes we discussed above, we believe that training supervisors in teaching skills alone is not a strong enough strategy to change the management practices of learners in difficulty within an institution.

Many recommendations have been published in the literature in recent years regarding the strategies to develop and maintain an institutional remediation program [14]. Lacasse et al published a BEME review providing a repertoire of literature-based interventions for helping learners experiencing academic difficulties [48]; Cleland et al. highlighted the importance for institutions to support learners in an educational alliance, and point to the need to consider how best to provide a supportive environment for all learners from a range of educational and social backgrounds to progress towards the required outcomes [49].

Our results are in line with these recommendations but add two perspectives that are also important to consider: the first in regard to the need to identify and fully understand not only the beliefs and the implicit norms, but also the feeling of self-efficacy that are shared by teachers and that are likely to motivate them to engage (or not) in the management of students in

difficulty. The second emphasizes the importance of analyzing these elements in the perspective of a change in practices and of identifying which factors may favor or hinder the change. This can only be done by analyzing the context, in close contact with the heads of the institutions concerned.

## Strengths and limitations

One of the major limitations of the study is the limited number of focus groups (FG). But the choice of our methodology remains consistent with the principles of action research, which is to involve people from the field who are interested in reflecting on a given issue with a view to improving processes. Another limitation is the monocentric approach to the study which may limit the generalization of the results; indeed the focus made in the context of Switzerland does come with own cultural, but also organizational differences. For example, our local context and the local norms rather reflect a culture of benevolence in supervision, but this questioning may nevertheless be generalized if we think of the issues relating to the best way to give a feedback in the literature [50].

Moreover, the faculty of medicine is not the governing body of the physicians once they have graduated as they are under another governing body which is the Swiss Institute for Advanced Medical Training and Education (SIWF). Keeping that in mind, the concerns raised by this study regarding the management of residents in difficulty in a Swiss general internal medicine outpatient clinic, is one that concerns most hospitals. Furthermore, the process of changing practices that we wanted to initiate through our chosen conceptual frameworks may also be relevant in other contexts.

Our methodological qualitative approach added value in that it facilitated a rich exploration of perceptions and beliefs. One major strength of our study is to have been able to "take the pulse" of a large department, and to have been able to create conditions which made it possible to make explicit the issues relating to the supervision of interns in difficulty as well as the difficulties encountered by their various supervisors, in order to then be able to initiate changes.

## Conclusion and perspectives

Participatory action research is a collaborative development process carried out by a group of people interested in changing existing practices in their context. The participants are then involved in a research process that looks at their current practices from a practical perspective [24].

According to Baum et al. [51], this action research process is carried out through a cycle of reflection, in which participants also become partners in the research process, including in deciding what action should stem from the research findings. Their opinion is particularly valuable given that they are involved in the study environment and that their common goal is the improvement of their activities.

This research process has fostered awareness and discussions at different levels. Results were shared, not only within the Primary Care Division, but also at a faculty management level. Since then, various actions and processes have been put in place at the Faculty of Medicine in Geneva. Amongst the most important is the appointment of a task force to develop and implement an institutional remediation program for the entire medical faculty at the pre-graduate level [52]. This makes it possible, particularly to identify learners in difficulty earlier in the course of their training. Discussions are underway for the post-graduate level. In addition, a training course consisting of several workshops and coaching over 2 years has been set up for all Chief residents. Some of the Primary care Division's Chief residents and attending physicians are involved in this training process. After having started in 2015 in five departments, it

will be extended in 2021 to all the departments of the Geneva University Hospitals receiving interns [53]. This training will also benefit students, since it is the same Chief residents who supervise them.

## Acknowledgments

We would like to warmly thank all the doctors who participated in the study. We also would like to thank Mrs. Julia Sader, a native English speaker and doctoral student at UDREM (Unit of Development and Research in Medical Education), for her careful proofreading and contribution to the manuscript.

## Author Contributions

**Conceptualization:** Cédric Lanier, Virginie Muller-Juge, Melissa Dominicé Dao, Noëlle Junod Perron, Marie-Claude Audétat.

**Formal analysis:** Cédric Lanier, Virginie Muller-Juge, Marie-Claude Audétat.

**Funding acquisition:** Cédric Lanier.

**Investigation:** Cédric Lanier.

**Methodology:** Cédric Lanier, Marie-Claude Audétat.

**Project administration:** Virginie Muller-Juge.

**Writing – original draft:** Cédric Lanier.

**Writing – review & editing:** Cédric Lanier, Virginie Muller-Juge, Melissa Dominicé Dao, Jean-Michel Gaspoz, Noëlle Junod Perron, Marie-Claude Audétat.

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
