## [Decision Letter · Decision Letter 0]

26 Oct 2020

PONE-D-20-15571

Management of residents in difficulty in a Swiss general internal medicine outpatient clinic: changes are needed!

PLOS ONE

Dear Dr. Cédric,

Thank you for submitting your manuscript to PLOS ONE. Sincere apologies for the delay in completing the initial round of the peer-review process. Twelve potential reviewers declined to review the manuscript citing various reasons. Finally, two reviewers have now submitted their comments.

After careful consideration, we feel that your manuscript has merit but does not fully meet PLOS ONE’s publication criteria as it currently stands. Therefore, we invite you to submit a revised version of the manuscript that addresses the points raised during the review process.

We look forward to receiving your revised manuscript.

Kind regards,

Ritesh G. Menezes, M.B.B.S., M.D., Diplomate N.B.

Academic Editor

PLOS ONE

Additional Comments by the assigned Academic Editor and Dr. Carmen Melatti, Associate Editor:

The manuscript reports qualitative research conducted in 2016. Are the results still relevant in 2020? Revise the manuscript accordingly.

The Swiss system would have adopted new measures for the management of residents in difficulty in the intervening time.

However, we also note that the qualitative methods uncovered findings that could be generalized to other settings (such as in general, the attitudes of mentors towards their residents, and their understanding of their role; and the resident's perceptions of failure), and that could be worthy of publication.

Discuss the findings in the setting of 2020: explaining what changes have been performed in the years since the interviews were carried out, and how their results can be generalized and used in the present setting.

Besides, address the comments/suggestions made by the reviewers.

Journal Requirements:

2. When reporting the results of qualitative research, we suggest consulting the COREQ guidelines: http://intqhc.oxfordjournals.org/content/19/6/349. In this case, please consider including more information on the number of interviewers, their training and characteristics; how participants were selected; if a pilot study was tested; how data was coded; if bias issues were considered.Moreover, please consider providing the interview guide as a supplementary material.

4. Thank you for stating the following in the Financial Disclosure Section of your manuscript:

"This project was supported by institutional Funding Elie SAFRA from the Faculty of Medicine,

University of Geneva."

We note that one or more of the authors are employed by a commercial company: Hirslanden Clinique des Grangettes.

5.1. Please provide an amended Funding Statement declaring this commercial affiliation, as well as a statement regarding the Role of Funders in your study. If the funding organization did not play a role in the study design, data collection and analysis, decision to publish, or preparation of the manuscript and only provided financial support in the form of authors' salaries and/or research materials, please review your statements relating to the author contributions, and ensure you have specifically and accurately indicated the role(s) that these authors had in your study. You can update author roles in the Author Contributions section of the online submission form.

5.2. Please also provide an updated Competing Interests Statement declaring this commercial affiliation along with any other relevant declarations relating to employment, consultancy, patents, products in development, or marketed products, etc.  

6. We note that you have indicated that data from this study are available upon request. PLOS only allows data to be available upon request if there are legal or ethical restrictions on sharing data publicly. For information on unacceptable data access restrictions, please see http://journals.plos.org/plosone/s/data-availability#loc-unacceptable-data-access-restrictions.

7. PLOS requires an ORCID iD for the corresponding author in Editorial Manager on papers submitted after December 6th, 2016. Please ensure that you have an ORCID iD and that it is validated in Editorial Manager. To do this, go to ‘Update my Information’ (in the upper left-hand corner of the main menu), and click on the Fetch/Validate link next to the ORCID field. This will take you to the ORCID site and allow you to create a new iD or authenticate a pre-existing iD in Editorial Manager. Please see the following video for instructions on linking an ORCID iD to your Editorial Manager account: https://www.youtube.com/watch?v=_xcclfuvtxQ

Reviewers' comments:

Reviewer's Responses to Questions

**Comments to the Author**

1. Is the manuscript technically sound, and do the data support the conclusions?

Reviewer #1: Yes

Reviewer #2: Partly

2. Has the statistical analysis been performed appropriately and rigorously? 

Reviewer #1: N/A

Reviewer #2: N/A

3. Have the authors made all data underlying the findings in their manuscript fully available?

Reviewer #1: Yes

Reviewer #2: Yes

4. Is the manuscript presented in an intelligible fashion and written in standard English?

Reviewer #1: No

Reviewer #2: Yes

5. Review Comments to the Author

Reviewer #1: I read the manuscript with great interest as this is a problem faced by many residency programs and there is a need to identify and establish a clear process for residents in difficulty. The manuscript is well-presented; however, a few issues need the attention of the authors.

1. Clarify what is meant by senior resident and how they are different from residents. Also please elaborate on their responsibilities regarding the supervision of residents.

2. Elaborate on how the current system identifies residents in difficulty, as it is mentioned that heads of divisions rarely establish the insufficient skills of residents.

3. In many of the categories, the supervisors’ views and opinions were well-presented; however, the resident’s views were limited. Particularly what residents thought of their supervisors’ abilities and skills in managing residents in difficulty.

4. The units for the information in Table 1 are not clear (e.g. supervision experience and clinical experience). Clarify what is meant by postgraduate title.

5. The overall structure and organization of the manuscript should be revised to make reading easier. For example, the part of ethical consideration should be moved.

6. There are issues with formatting that need revision. The legends for figures are found within the body of the manuscript (not the figures themselves). The information written in Box1 can be included in the body of the manuscript and not as a box.

7. The manuscript would benefit from editing the grammar and punctuation.

Reviewer #2: This is an interesting article about an very actual topic in postgraduate medical education. Using the Fishbein's framework authors describe the different influencing (facilitating and inhibiting) factors involved in the supervision and management of residents in difficulty.

The article is well written and easy readable.

The article offert a good framework to setup concrete actions.

I have some experience in qualitative research but not at all with the design of Participation (or participatory?) action research nor with the immersion /crystallisation data analysis, so the following comments my be not entirely appropriate. The authors may consider to adress my suggestions or not.

Concerning the study design:

The purpose of Participatory action research is to enable action. According to Baum et al. "Action is achieved through a reflective cycle, whereby participants collect and analyse data, then determine what action should follow. The resultant action is then further researched and an iterative reflective cycle perpetuates data collection, reflection, and action as in a corkscrew action......The researched cease to be objects and become partners in the whole research process: including selecting the research topic, data collection, and analysis and deciding what action should happen as a result of the research findings." (Journal of Epidemiology & Community Health 2006;60:854-857 http://dx.doi.org/10.1136/jech.2004.028662).

In the actual study the iterative reflection cycle and the involvement of the researched objectives (supervisors of residents in difficulty) is not described.  could you comment on it?

Data collection:

one of the major limitation of the study is the limited number of focus groups (FG) and the single center context of the stydy that may limit generalization of results  add a praragraph about limitations

It may not be clear for the reader why the Chief physician was not involved in FG with attending physicians. What was the aim and the added value to interview separately the Chief physician. Do questions to him differ from other groups?

FG was runned with 6 or 7 participants, which is good. It could be interesting to know how many physicians from the primary care division were eligible to participate to the FG and how the recruitment was performed (do they have volunteered to participate or or a subset of them was recruited?) How many refused to participate or were not invited?

CAVE: about the number of participants to FG the text say 6 residents bur the table 1 state 7 residents please correct

FG last about 80 minutes and in these 80 min 7 participants were exposed to 9 questions: hence each participant had a mean of 1.5 minutes to answer, discuss and deepen, counter opinions for other participants for each point, which seem ambitous to an in depth discussion  could you comment on this ?

Results:

Presentation and organisation of results are adequate and figure 2 resume well grahically the main results

Discussion:

At the end of the first paragraph "Summary of results" you say that your results are in line with specialized literature, without specifing what parts of your results are similar and to which articles you refer. Do you mean other aticles using the Fishbein's framework or also studies that used other framework to investigate the problem?

It would be interesting to put results more in perspective with the existing literature and the results of other studies. Are there elements that are different from what other groups found?

The local context and the small number of FG raise the question about the generalisability of results outside western Switzerland. How do your results about Attitude and behavioural beliefs, normsand normative belief and self-efficacy and efficacy belief relate to other studies. They are probably related to the local context ans culture  please comment on it

The study has some limitations (e.g. number of participants, number of FG, monocentric design) that limit the generalisability of results. I suggest Limitations to be adressed in a specific paragraph.

Specific comments:

In result section correct the different number of participants in residents FG between text and tsable 1

End of the first paragraphe of Discussion: change "but our in depth analyze permits..." to "but our analysis allows..."

6. PLOS authors have the option to publish the peer review history of their article (what does this mean?). If published, this will include your full peer review and any attached files.

Reviewer #1: No

Reviewer #2: No

---

## [Author Response · Author response to Decision Letter 0]

14 May 2021

Dear Dr. Ritesh G. Menezes, 

Academic Editor PLOS ONE

Thank you for giving us the opportunity to submit a revised version of our manuscript “Management of residents in difficulty in a Swiss general internal medicine outpatient clinic: changes are needed!” (ref.no. PONE-D-20-15571) to “PLOS ONE”. We appreciate the time and effort that you and the Reviewers have dedicated to providing your valuable feedback on our manuscript, and we are grateful for your comments on our paper. All suggestions have been incorporated. Here is an itemized, point-by point response to each comment and concern.

---

## [Decision Letter · Decision Letter 1]

16 Jun 2021

PONE-D-20-15571R1

Management of residents in difficulty in a Swiss general internal medicine outpatient clinic: changes are needed!

PLOS ONE

Dear Dr. Cédric,

Thank you for submitting your manuscript to PLOS ONE. After careful consideration, we feel that it has merit but does not fully meet PLOS ONE’s publication criteria as it currently stands. Therefore, we invite you to submit a revised version of the manuscript that addresses the points raised during the review process.

Please submit your revised manuscript by 22-June-2021. Please include the following items when submitting your revised manuscript:

We look forward to receiving your revised manuscript.

Kind regards,

Prof. Ritesh G. Menezes, M.B.B.S., M.D., Diplomate N.B.

Academic Editor

PLOS ONE

Journal Requirements:

Additional Academic Editor Comments:

- Some errors of English grammar and use still persist in the revised manuscript.

Reviewers' comments:

Reviewer's Responses to Questions

**Comments to the Author**

1. If the authors have adequately addressed your comments raised in a previous round of review and you feel that this manuscript is now acceptable for publication, you may indicate that here to bypass the “Comments to the Author” section, enter your conflict of interest statement in the “Confidential to Editor” section, and submit your "Accept" recommendation.

Reviewer #1: All comments have been addressed

Reviewer #2: All comments have been addressed

2. Is the manuscript technically sound, and do the data support the conclusions?

Reviewer #1: Yes

Reviewer #2: Yes

3. Has the statistical analysis been performed appropriately and rigorously? 

Reviewer #1: Yes

Reviewer #2: N/A

4. Have the authors made all data underlying the findings in their manuscript fully available?

Reviewer #1: Yes

Reviewer #2: Yes

5. Is the manuscript presented in an intelligible fashion and written in standard English?

Reviewer #1: No

Reviewer #2: Yes

6. Review Comments to the Author

Reviewer #1: Thank you for your response. I enjoyed reading the revised manuscript and I see you addressed the majority of the issues that were raised previously.

There are still minor issues with punctuation and some word choices (e.g. beneficiate line 16) that would benefit from revision.

Reviewer #2: Authors have answered in a satisfactory way to all comments. The articulation of sentences and concepts is more fluid. I have no additional comments

7. PLOS authors have the option to publish the peer review history of their article (what does this mean?). If published, this will include your full peer review and any attached files.

Reviewer #1: No

Reviewer #2: No

---

## [Author Response · Author response to Decision Letter 1]

21 Jun 2021

Reviewer 1 

Minor concerns: 

Thank you for your response. I enjoyed reading the revised manuscript and I see you addressed the majority of the issues that were raised previously.

There are still minor issues with punctuation and some word choices (e.g. beneficiate line 16) that would benefit from revision. Thank you for your feedback, we are delighted that the revised manuscript addressed most of the issues you raised. 

Answer: We have decided to ask an English native to proofread our manuscript and made quite a few changes. We thank you for the suggestion and believe that the changes will clear this minor issue and that is has strengthened the clarity and fluidity of our manuscript. Please see manuscript for a detailed look at the changes made. 

Reviewer 2 

Authors have answered in a satisfactory way to all comments. The articulation of sentences and concepts is more fluid. I have no additional comments 

Answer: Thank you for your feedback, it is nice to hear that you are satisfied with our responses. Please see manuscript for a detailed look at the changes made. 

Additional Academic Editor Comments: 

Some errors of English grammar and use still persist in the revised manuscript. 

Answer: Thank you for your general comment, as mentioned before we have addressed these concerns and ask an English native speaker to proofread our manuscript. 

Thank you for the opportunity to publish in your journal and we hope that these last changes strengthened our manuscript and that you will be as happy with its content as we are now. Please see manuscript for a detailed look at the changes made.

---

## [Editor Report · Decision Letter 2]

28 Jun 2021

Management of residents in difficulty in a Swiss general internal medicine outpatient clinic: change is necessary!

PONE-D-20-15571R2

Dear Dr. Cédric,

We’re pleased to inform you that your manuscript has been judged scientifically suitable for publication and will be formally accepted for publication once it meets all outstanding technical requirements.

Kind regards,

Prof. Ritesh G. Menezes, M.B.B.S., M.D., Diplomate N.B.

Academic Editor

PLOS ONE

---

## [Editor Report · Acceptance letter]

7 Jul 2021

PONE-D-20-15571R2 

Management of residents in difficulty in a Swiss general internal medicine outpatient clinic: change is necessary! 

Dear Dr. Lanier:

I'm pleased to inform you that your manuscript has been deemed suitable for publication in PLOS ONE. Congratulations! Your manuscript is now with our production department. 

Kind regards, 

on behalf of

Prof. Dr. Ritesh G. Menezes 

Academic Editor

PLOS ONE